# Efficiently Serving Large Multimodal Models Using EPD Disaggregation

**Gursimran Singh** [1]  **Xinglu Wang** [2]  **Yifan Hu** [1]  **Timothy Yu** [1]  **Linzi Xing** [1]  **Wei Jiang** [3]  **Zhefeng Wang** [3]
**Xiaolong Bai** [3]  **Yi Li** [3]  **Ying Xiong** [1]  **Yong Zhang** [1]  **Zhenan Fan** [1]

## Abstract

Large Multimodal Models (LMMs) extend Large Language Models (LLMs) by handling diverse inputs such as images, audio, and video, but at the cost of adding a multimodal encoding stage that increases both computational and memory overhead. This step negatively affects key Service Level Objectives (SLOs), such as time to first token (TTFT) and time per output token (TPOT). We introduce Encode-Prefill-Decode (EPD) Disaggregation, a novel framework that separates the encoding, prefill, and decode stages onto dedicated resources. Unlike current systems, which bundle encoding and prefill together, our approach decouples these steps, unlocking new opportunities and optimizations. These include a mechanism to cache multimedia tokens for efficient transfer, a novel way to parallelize the encoding load within a request, a module for optimal resource allocation for disaggregated serving, and a novel role-switching method to handle changing workload characteristics. Experimental evaluations with popular LMMs show substantial gains in memory efficiency (up to 15× lower peak memory utilization), batch sizes (up to 22× larger), 10× more images per request, and 2.2× larger KV caches. Furthermore, it leads to significant improvements in SLO attainment (up to 90–100% improvement) and TTFT (up to 71% reduction), compared to systems that do not disaggregate. The code is available at https://github.com/vbdi/epdserve.

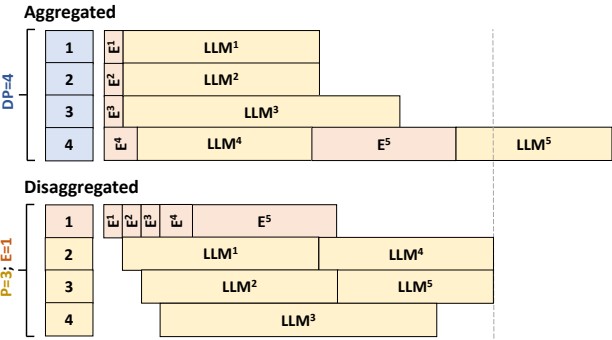

Figure 1: Aggregated (top) vs. disaggregated (bottom) system architectures. In the aggregated setup, the encoder (E) and LLM share the same GPUs, leading to interference between encode and prefill stages (e.g., LLM-4 delays E5). Disaggregation isolates these stages across GPUs, reducing contention and improving utilization. This separation enables better performance under multimodal workloads, addressing key limitations of existing systems.

## 1. Introduction

Large Language Models (LLMs) have revolutionized language understanding and reasoning, achieving superhuman performance on a variety of tasks (Achiam et al., 2023; Chang et al., 2024). Recently, the scope of these models has expanded to include multiple modalities, such as images, audio, and videos, leading to the emergence of Large Multimodal Models (LMMs) (Yao et al., 2024; Liu et al., 2023; Chen et al., 2024). LMMs enable users to interact with diverse data types, such as posing questions about visual scenes or analyzing audio clips, thereby unlocking novel applications across fields like healthcare, autonomous systems, and creative industries (Luo et al., 2025).

However, serving LMMs in an efficient manner presents unique challenges. Meeting strict Service Level Objectives (SLOs), such as time to first token (TTFT) and time per output token (TPOT), becomes increasingly difficult given the added computational and memory demands of processing multimodal data (Alvar et al., 2025; Liu et al., 2024). Unlike LLMs, where inference involves prefill and decoding stages, LMMs require an additional encoding stage to process raw multimodal inputs (e.g., images or videos) into tokenized

[1] Huawei Technologies Canada, BC, Canada [2] Simon Fraser University, BC, Canada [3] Huawei Cloud, China. Correspondence to: Zhenan Fan < zhenanfan@gmail.com >, Gursirman Singh <gursimran.singh1@huawei.com>.

*Proceedings of the $42^{nd}$ International Conference on Machine Learning*, Vancouver, Canada. PMLR 267, 2025. Copyright 2025 by the author(s).

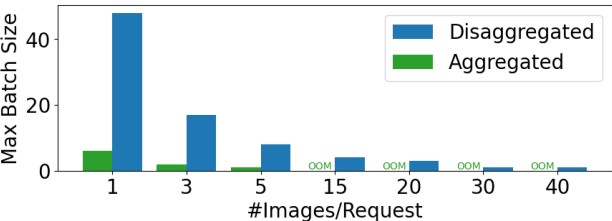

Figure 2: Impact of disaggregation on supported batch size and number of images per request for the MiniCPM-V 2.6 model. Removing the LLM from the GPU significantly increases capacity, enabling larger batches and higher-resolution inputs. This demonstrates the memory efficiency benefits of disaggregation.

representations. This stage is computationally intensive, especially for high-resolution or complex multimodal inputs, and often produces a substantial number of additional tokens (Wu et al., 2023). The resulting token inflation increases resource consumption and leads to quadratic growth in prefill-stage compute demands, adversely impacting SLO attainment.

Disaggregating the prefill stage from the decode stage has emerged as a well-studied solution for improving LLM inference efficiency (Zhong et al., 2024; Qin et al., 2024; Patel et al., 2024; Jin et al., 2024; Hu et al., 2024). By assigning separate resources to each stage, prefill-decode disaggregation enables independent optimization of batching, scheduling, and resource allocation strategies, significantly enhancing system throughput and memory utilization. However, these techniques fall short in addressing LMM-specific challenges, as the addition of an encoding stage fundamentally changes the resource dynamics. The encoding stage adds significant compute and memory overhead, inflates tokens, and creates dependencies that affect later stages, requiring a fresh look at disaggregation strategies for multimodal workloads.

These challenges present optimization opportunities that current serving systems do not exploit. Presently, the encode and prefill stages are aggregated into a single monolithic and synchronous step executed on the same set of GPUs. Sequential execution of encoding and prefilling introduces interference, as demonstrated in Figure 1. In the aggregated setup (top), the prefill step ($LLM^4$) interferes with the encode-heavy request ($E^5$). As a result, such aggregation leads to suboptimal resource utilization and degraded SLO performance, revealing the inadequacy of current solutions for LMM workloads. Disaggregating encoding from prefill enables the system to reduce such interference under specific workloads. From another perspective, simultaneously loading both the multimodal encoder (MME) and the LLM onto the same GPUs in aggregated setups restricts the available memory for processing multiple high-resolution

images or supporting larger batch sizes. In preliminary investigations, removing the LLM and its associated KV cache significantly increases the maximum batch size and the number of images per request (Figure 2).

Thus, we propose Encode-Prefill-Decode (EPD) Disaggregation, a framework that decouples the encode, prefill, and decode stages, assigning each stage dedicated resources to operate independently. This design enables customized strategies for batching, parallelization, and scheduling at each stage, optimizing resource utilization while reducing contention. As a result, EPD achieves better memory utilization, higher throughput, and improved compliance with critical SLOs like TTFT and TPOT.

The major contributions of this work are as follows:
- We propose an efficient system for LMM inference that introduces the novel idea of disaggregating the encoding and prefill stages. Our approach addresses key challenges in inter-stage communication, efficient parallelization, resource allocation, and performance optimization within this framework.
- We introduce intra-request parallelization (IRP), which shards a request into independent encoding jobs that can be executed in parallel, significantly reducing first-token latency.
- We formulate the resource allocation problem as an optimization over batch sizes, scheduling strategies, and parallelization approaches for each pipeline stage. Furthermore, we provide a simple black-box optimizer that uses workload samples to approximate the optimal configuration for a given workload.
- We develop a dynamic role-switching capability that monitors the system for bottlenecks and enables flexible reallocation of resources across the E, P, and D stages by switching an instance's role between stages. This ensures the system can respond effectively to changes in workload requirements during online serving.
- We conduct evaluations on several popular LMM models—MiniCPM-V 2.6, InternVL2-8B, and InternVL2-26B—using both real-world and synthetic workloads. Results demonstrate the superiority of our approach: up to 15× lower memory usage, 22× larger batch sizes, 10× more images per request, up to a 90–100% improvement in SLO attainment, and up to 71% lower TTFT.

## 2. Related Work

In this section, we describe existing approaches in the literature for multimodal model serving and their limitations, and then we review the use of disaggregation techniques.

**Multimodal Model Serving.** There are two ways to enable a multimodal model serving system: (1) adopting and extending existing LLM serving systems, such as vLLM

(Kwon et al., 2023), SARATHI (Agrawal et al., 2023), and Orca (Yu et al., 2022); and (2) leveraging open-source code from recent works on improving multimodal model inference. The first approach fails to address the encoding bottleneck inherent in LMMs, limiting their applicability to multimodal workloads, particularly when handling high-volume and high-resolution multimedia data. In the second approach, recent advancements such as KV cache eviction (Li et al., 2024; Ning et al., 2024) and compression (Kang et al., 2024) focus on specific features and limited scenarios. For instance, Inf-MLLM (Ning et al., 2024) enables efficient streaming inference for LMMs on a single GPU, targeting resource-constrained scenarios. While these implementations are feasible for serving LMMs, they fail to meet user SLOs in cloud serving scenarios. To the best of our knowledge, we are the first to propose an LMM serving system that leverages disaggregation and a series of integrated techniques to enable better resource allocation and improved SLOs.

**Disaggregated Serving.** Disaggregated serving has emerged as a promising technique in large model serving. By decoupling the prefill and decode stages, systems like SplitWise (Patel et al., 2024), DistServe (Zhong et al., 2024), and DéjàVu (Strati et al., 2024) mitigate interference between these phases, enabling finer control over TTFT and TPOT. However, these systems primarily target LLMs and overlook the encoding step required for LMMs, which is tightly coupled with prefill. Recent work, such as Mooncake (Qin et al., 2024) and PD-Serve (Jin et al., 2024), have extended disaggregation to include KV cache management and other system-level optimizations. Despite these advancements, they remain limited in their applicability to LMMs. Disaggregating the encoding phase from prefill could unlock new opportunities for optimizing LMM serving. For instance, when handling requests with long videos, the frames could be encoded in parallel, reducing head-of-line blocking and improving overall system efficiency. This approach would enable more flexible batching and scheduling strategies, achieving better latency and throughput for multimodal workloads.

# 3. Method

In this section, we first describe the EPD disaggregation framework (Section 3.1), followed by a detailed explanation of the system design and optimizations (Section 3.2), including asynchronous token transfer, intra-request parallelism, optimized resource allocation, and dynamic role switching.

## 3.1. EPD Disaggregation

As shown in Figure 3, disaggregating an LMM system involves dividing the inference process into three stages: *encoding*, *prefill*, and *decode*. Transitions between these

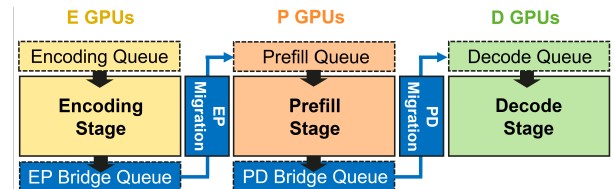

Figure 3: The inference pipeline of EPD Disaggregation.

stages—*EP-migration* and *PD-migration*—handle the transfer of data from encoding to prefill and from prefill to decode, respectively. We denote the input text prompt as $i_p$, multimodal data as $i_m$, and the output text as $o$. The steps are as follows:

**Encoding (E):** The multimodal input $i_m$ is processed by the multimodal encoder $E$, converting it into multimodal tokens $v_t^e = E(i_m)$, which form a high-dimensional embedding for the next pipeline stage.

**EP-migration:** Once encoding completes, the generated tokens are transferred to the prefill stage $P$ via the *EP-migration* function $\psi_{EP}$, such that $v_t^p = \psi_{EP}(v_t^e)$.

**Prefill (P):** The prefill stage processes $v_t^p$ and the text prompt $i_p$ to produce the initial KV cache and first output token: $kv_1^p, o_1^p = P(v_t^p, i_p)$.

**PD-migration:** The KV cache and token are passed to the decode stage via the *PD-migration* function $\psi_{PD}$, where $kv_1^d, o_1^d = \psi_{PD}(kv_1^p, o_1^p)$.

**Decode (D):** Decoding proceeds autoregressively, generating the next token $o_{t+1}^d$ and updating the cache: $kv_{t+1}^d, o_{t+1}^d = D(kv_t^d, o_t^d)$.

This autoregressive process continues until the output sequence is fully generated.

## 3.2. System Design and Optimization

Figure 4 illustrates the architecture of the EPD Disaggregated inference system. Each pipeline stage (Encoding, Prefill, and Decoding) has independent instances that run the corresponding stage. These instances operate in data parallel (DP) mode, enabling concurrent processing of multiple requests per stage and ensuring scalability and efficiency.

Each instance comprises a scheduler, responsible for scheduling requests, block managers (responsible for managing cache(s)), and multiple workers. The workers operate in tensor-parallel (TP) and/or pipeline-parallel (PP) mode, where each worker holds only a subset of the model weights and the corresponding caches required for the stage.

In the encoding stage, workers load only the encoder weights and initialize the Multimodal (MM) cache. In the prefill stage, the LLM weights are loaded, and both the MM and KV caches are required to efficiently manage all

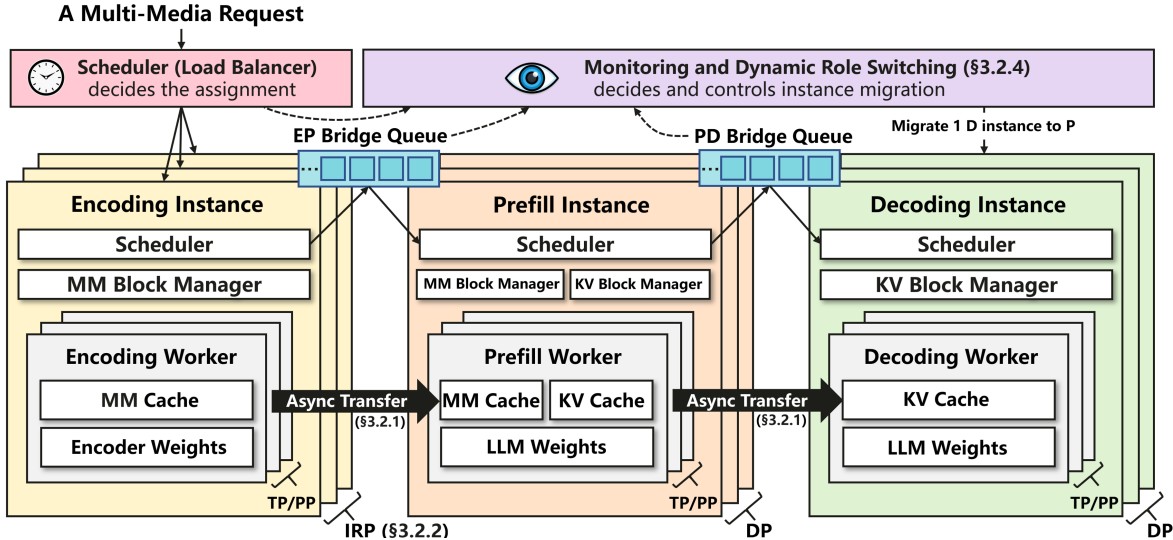

Figure 4: System architecture of the proposed EPD Disaggregated Inference.

the data associated with the request. In the decoding stage, workers load the LLM weights for decoding tasks and use the KV cache.

Cache transfers occur asynchronously when an instance pulls a request from its queue, ensuring the transfer occurs once the downstream instance is ready. Further, our system incorporates several techniques to optimize the performance of the disaggregated LMM pipeline, with a primary focus on ensuring smooth token transfers between stages (Section 3.2.1), reducing latency (Section 3.2.2), managing resources effectively (Section 3.2.3), and adapting to changing workloads (Section 3.2.4). The ablation of these features is shown in Section 4.4.

### 3.2.1. ASYNCHRONOUS TOKEN TRANSFER

Disaggregating the system introduces an additional step of transferring vision tokens from the encoding to the prefill stage. To minimize latency during token transfers between stages, our system employs direct, asynchronous transfers via high-bandwidth channels (NVLink, and InfiniBand). The asynchronous transfer allows the system to continue processing new requests without interruption. Both encoding and prefill workers maintain an MM cache to facilitate this process.

When encoding is complete, tokens are stored in the encoding worker's MM cache, allowing it to serve new requests immediately. An asynchronous event loop monitors completed encoding tasks and initiates direct token transfers to the prefill worker's MM cache. Once the transfer is confirmed, the encoding cache entries are cleared to free memory. To manage these cache blocks effectively, we introduce the MMBlockManager, which pre-allocates cache blocks based on each request's needs. After token transfer,

the blocks are reassigned or de-allocated, ensuring flexible cache utilization even under heavy workloads.

### 3.2.2. INTRA-REQUEST PARALLEL (IRP)

Multimodal requests often include multiple high-resolution images in many practical scenarios like autonomous driving and video question answering. In modern LMMs, these images are further converted into a large number of patches that need to be processed through a computationally heavy MME. This significantly increases the computation load, making the encoding process a major bottleneck.

To address this, we introduce Intra-Request Parallelism (IRP), which partitions a single request's image patches across multiple encoding workers in a data-parallel fashion. Since patches are encoded independently, they can be processed and transferred concurrently. Specifically, each encoding worker concurrently processes a subset of patches, computes their token representations, and asynchronously transfers them to the prefill stage. Once all patch-level tokens reach the prefill stage, they are aligned, projected, and merged to form the complete multimodal tokens.

### 3.2.3. OPTIMIZED RESOURCE ALLOCATION

The EPD system requires tuning various configurations, including batch sizes, scheduling strategies, and parallelization methods for each pipeline stage. To determine the optimal setup, we collect historical workload samples and apply a black-box optimizer.

Let $f(\cdot)$ represent the system's performance metric (e.g., goodput, see Section 4). Since $f(\cdot)$ is treated as a black-box function with an unknown internal mechanism, we rely on a simulator—extended from DistServe (Zhong et al.,

2024)—to evaluate performance metrics efficiently. The objective is to maximize performance while minimizing GPU usage, which inherently reduces pipeline inefficiencies (e.g., idle time) and improves resource utilization. Formally, we solve

$$\max_{(\mathbf{p},\mathbf{b},\mathbf{s})\in\mathcal{X}} f(\mathbf{p},\mathbf{b},\mathbf{s}) - \beta cost(\mathbf{p}) \qquad (1)$$

Here, $\mathcal{X}$ is the search space for system configs, including parallelization configs $\mathbf{p}$, max batch size configs $\mathbf{b}$, and scheduling configs $\mathbf{s}$ (See Appendix D for details). We use Bayesian optimization (Calvo et al., 2019) to solve Problem 1.

### 3.2.4. DYNAMIC ROLE SWITCHING

The configuration optimizer described in the previous section can determine the optimal settings for a given workload. However, in an online environment, workload characteristics can change dynamically, requiring adjustments to the configuration. Re-initializing the entire system from scratch in response to these changes can be both difficult and costly. For instance, there may be ongoing requests in the encoding, context, or decoding phases with precomputed KV and/or VE caches stored in memory. A naive re-initialization would not only involve booting time but also force these requests to restart from the beginning, potentially causing a cascading impact on SLO metrics for subsequent requests.

To overcome these challenges, we introduce dynamic role switching, which enables any instance in the E, P, or D stages to switch roles to any other stage (E, P, or D) with minimal overhead. At a high level, dynamic role switching continuously monitors the system's queuing statistics across all stages and reallocates workers to stages experiencing higher demand. When a decision is made to transfer an instance from a source stage $S$ to a destination stage $T$, the migration process occurs in three key steps:

- Offload: The instance in the $S$ stage stops accepting new requests and redistributes its queued tasks to sibling instances in the same stage.
- Migration: The instance is reconfigured to meet the requirements of the $T$ stage. This may involve switching both the model and cache type. For example, if the E stage is involved, the instance may switch from an LLM to an MME model and from a KV cache to an MM cache.
- Onload: The migrated instance resumes processing new requests, helping to alleviate the queuing bottleneck at the $T$ stage.

This process typically takes less than 0.7 seconds. The duration is longer for migrations involving the E stage due to model and cache changes, but significantly shorter when switching between P and D stages, as both the LLM and KV cache can be reused.

## 4. Experiments

In this section, we analyze and compare the performance of the proposed EPD disaggregation method against various baselines. We start with an end-to-end generation performance analysis in Section 4.1, followed by an examination of the first token latency in Section 4.2. We then evaluate the memory savings achieved by EPD in Section 4.3. Next, we present an ablation analysis of the key system components in Section 4.4. Finally, we present an extension of our framework to Neural Processing Units (NPUs) in Section 4.5. Additional experiments are detailed in the Appendix, including 1) a throughput comparison for offline and heterogeneous settings in Appendix A.3, 2) additional analysis of the implementation on NPUs in Appendix F, 3) an experiment extending our framework to the audio modality in Appendix A.1, and 4) additional SLO and memory experiments in Appendix A.

**Baselines:** We compared the proposed EPD method against two popular baselines: DistServe (Zhong et al., 2024) and vLLM (Kwon et al., 2023). The DistServe baseline implements the prefill-decode (PD) disaggregation approach, in which the prefill and encoding phases are executed on one set of GPUs, while the decode phase is disaggregated on separate GPUs. Since DistServe was originally designed for LLMs, we extended it to support LMMs by enabling multimodal data processing and modifying its block manager to accommodate multimodal tokens. The vLLM baseline adopts a monolithic architecture, where all three stages run on the same set of GPUs.

**Models:** We utilized three LMMs in our analysis: MiniCPM-V 2.6 (Yao et al., 2024), InternVL2-8B, and InternVL2-26B (Chen et al., 2024). These LMMs are renowned for their advanced capabilities in processing and understanding multimodal data. Detailed descriptions of the LMMs and their sizes can be found in Appendix E.2.

**Datasets:** To evaluate performance across diverse scenarios, we use three datasets: synthetic workload, NextQA, and Video-MME. The *synthetic workload* enables configurable parameters such as prompt length, number of images per request, image resolution, output length, and sampling settings. Unless otherwise noted, the input prompt length is set to 22 tokens. *NextQA* (Xiao et al., 2021), a benchmark video question-answering dataset, features human-annotated questions and answers, offering a more realistic reflection of real-world video request distributions compared to the synthetic workload. *Video-MME* (Fu et al., 2024) is a multimodal evaluation dataset designed for assessing large LMMs on video understanding tasks. It contains multiple-choice video

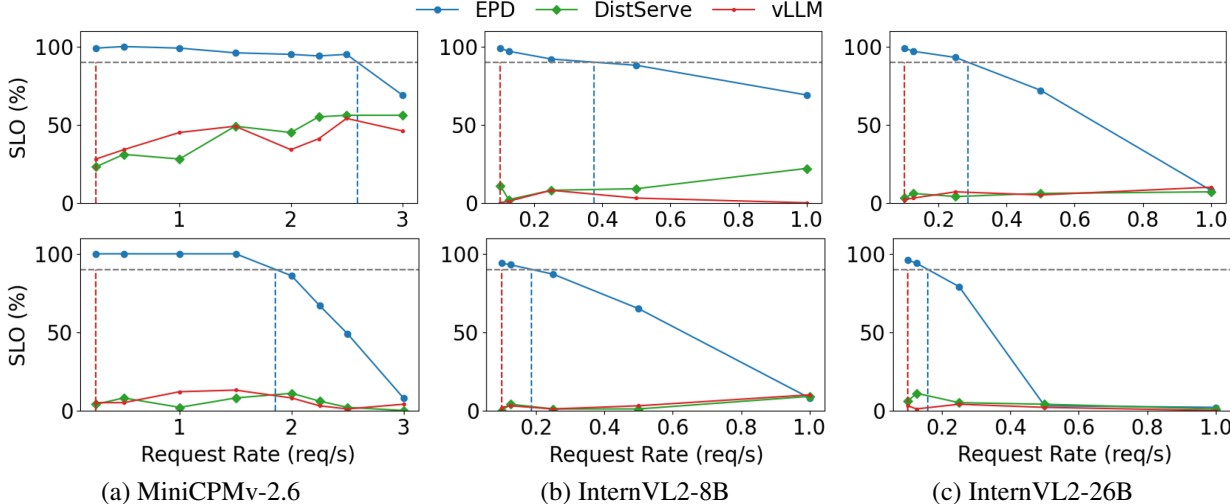

Figure 5: SLO attainment (↑) for end-to-end inference across multiple models and image counts per request. Subfigures (a), (b), and (c) correspond to MiniCPM-V 2.6, InternVL2-8B, and InternVL2-26B, respectively. The top and bottom rows show results for 2 and 4 images per request. EPD consistently outperforms all baselines across configurations.

question answering items that span diverse video lengths, topics, and reasoning types. In contrast to NextQA, which emphasizes open-ended video-text generation, Video-MME focuses strictly on multiple-choice QA and covers a broader spectrum of video durations and formats.

**Evaluation Metrics:** We evaluate the system based on runtime performance and memory consumption. The performance metrics are as follows:

- TTFT: The time from the submission of a request to the system until the first token is received by the user.
- TPOT: The average time interval between consecutive output tokens (excluding the first token).
- SLO Attainment: The percentage of requests that meet predefined SLOs, such as TTFT and TPOT requirements.
- Goodput: The highest request rate at which 90% or more SLO attainment is achieved.

For memory benchmarking experiments, we analyze the baselines by evaluating the benefits of available free memory in each approach. Additional free memory can facilitate higher batch sizes, accommodate more images per request, or enable larger key-value (KV) cache sizes (see Section 4.3).

### 4.1. SLO Attainment for End-to-End Generation

In this experiment, we evaluate the goodput of EPD and the baselines in an online setting where 100 multimodal requests arrive following a Poisson process with rate $\lambda$. The length of output tokens is fixed to 10. We test requests with 2 and 4 images, each at a resolution of $4032 \times 3024$. The

details of the corresponding SLO criteria are discussed in Appendix E.3. The results for three models (columns) are presented in Figure 5. The X-axis represents the overall request rate ($\lambda$), and the Y-axis indicates the percentage of requests meeting both TTFT and TPOT requirements, with a 90% threshold denoted by a black dotted line.

EPD outperforms all baselines, achieving over 90% SLO attainment at lower request rates. This is due to its ability to parallelize the computationally intensive image encoding step across multiple GPUs. DistServe and vLLM often maintain less than 10% SLO attainment due to interference. Meanwhile, comparing the first and second rows for varying image counts per request (2 and 4), we observe that, while more images per request increase the workload, EPD maintains reasonable performance, whereas baselines' SLO attainment drops significantly. Lastly, InternVL, which is prefill-heavy, experiences queuing delays due to the higher number of image tokens, particularly at higher request rates. This effect further exacerbates the performance degradation from the 8B to the 26B model. In contrast, MiniCPM-V, optimized to generate fewer image tokens, avoids these delays and achieves lower overall latency. Further results with 6 and 8 images are provided in Appendix A.4.

Next, we repeat the experiment using the non-synthetic video question-answering dataset, NextQA (Xiao et al., 2021). To do so, we randomly sampled 100 examples, with input text token lengths ranging from 4 to 21 (average: 11.42) and output token lengths ranging from 1 to 7 (average: 2.75). Each video request was represented by 8 uniformly sampled frames, and we used MiniCPM-V 2.6 in the experiment, adhering to the SLO criteria of TTFT = 5.60 and TPOT = 0.06. As illustrated in Figure 7, EPD is

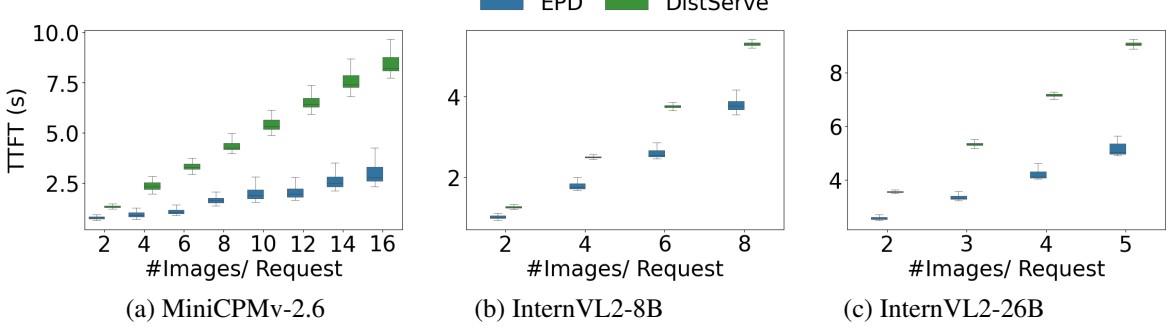

Figure 6: Distribution of TTFT (Y-axis) across varying numbers of images per request (X-axis) for (a) MiniCPM-V 2.6, (b) InternVL2-8B, and (c) InternVL2-26B. Each plot illustrates the latency behavior under increasing input sizes.

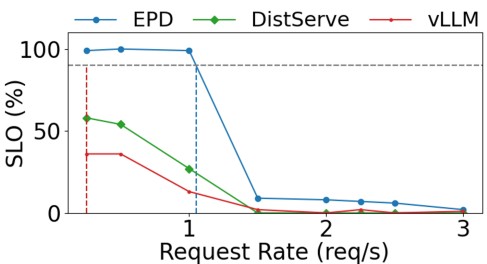

Figure 7: SLO attainment (↑) versus request rate on the NextQA dataset using the MiniCPM-V 2.6 model. EPD consistently achieves higher SLO attainment compared to all baselines.

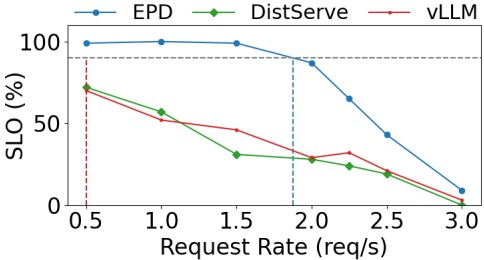

Figure 8: SLO attainment (↑) versus request rate on the Video-MME dataset using the MiniCPM-V 2.6 model. EPD significantly outperforms competing baselines across all request rates.

the only framework achieving 90% SLO attainment at low request rates, demonstrating its superior ability to handle real-world workloads compared to DistServe and vLLM.

Finally, we conduct the same experiment using the Video-MME (Fu et al., 2024) dataset. We evaluate SLO attainment, defined as TTFT $\leq$ 3.1s and TPOT $\leq$ 0.025s, on 100 randomly sampled Video-MME examples using MiniCPM-V 2.6. Each video is represented by 64 uniformly sampled frames, following the MiniCPM frame configuration reported on the Video-MME leaderboard. The results are shown in Figure 8.

As seen, EPD consistently outperforms vLLM and DistServe across all rates, demonstrating a strong generalization to temporal multimodal workloads, such as videos.

### 4.2. First Token Generation Latency

Multimodal requests often impose a heavy load on both the E and P phases, making the first token generation a key latency bottleneck. Therefore, in this experiment, we analyze the first token latency for various baselines. The results presented in Figure 6 show box plots of TTFT distributions for three different models. Note that, as the decoding phase is excluded, the vLLM baseline is equivalent to DistServe and is thus omitted. Requests are generated according to a Poisson distribution with a fixed request rate $\lambda$. Specifically, $\lambda = 0.25$ for MiniCPM-V 2.6 and $\lambda = 0.08$ for both InternVL2-8B and InternVL2-26B. Thanks to intra-request parallelization, EPD significantly outperforms both vLLM and DistServe. Specifically, TTFT is reduced by up to 71.9%, 32.8%, and 44.9% compared to the DistServe baseline for the MiniCPM-V 2.6, InternVL2-8B, and InternVL2-26B models, respectively.

Additionally, we conducted a TTFT comparison on the Video-MME (Fu et al., 2024) dataset for various baselines. The results are presented in Table 1. As seen, EPD achieves a latency reduction of up to 68.2% over DistServe and up to 69.2% over vLLM. Moreover, the performance gap widens with increasing video length—for instance, at 8 frames, EPD reduces latency by 42.9%, and at 64 frames, the reduction

| Method \ #Frames | 8 | 16 | 32 | 64 |
|---|---|---|---|---|
| vLLM | 0.42 | 0.82 | 1.59 | 3.11 |
| DistServe | 0.42 | 0.81 | 1.54 | 3.08 |
| EPD (ours) | *0.24* | *0.30* | *0.49* | *1.00* |

Table 1: Mean TTFT latency (in seconds) (↓) for varying video lengths at a fixed request rate of 1 request/sec. Results are averaged over 100 Video-MME samples. EPD achieves the lowest latency across all video lengths.

reaches 67.5% relative to DistServe. These results highlight EPD's superior scalability and robustness under increasingly demanding video processing workloads.

### 4.3. Memory Savings through Stage Disaggregation

In this section, we analyze the memory savings achieved by disaggregating the encoding and prefill stages. The E workers can save memory as they do not require LLM weights or the KV cache. Analyzing only the weight size indicates memory reduction of approximately 95%, 96.2%, and 78.3% for the MiniCPM-V 2.6, InternVL2-8B, and InternVL2-26B models. Similarly, for the P workers, memory savings of about 5%, 3.7%, and 21.6% are achieved, respectively. In practice, since KV cache is also not required at E workers, the memory saving can be even higher (93.3% saving, i.e., $15\times$ lower, according to profiling). These reductions in memory usage enable the EPD system to support higher numbers of images per request and batch sizes (discussed in the following experiments) and larger KV cache sizes (discussed in Appendix A.2).

| Model | Image Reso. | DistServe | EPD |
|---|---|---|---|
| MiniCPM-V 2.6 | 313,234 | 77 | *490* |
| | 787,444 | 26 | *165* |
| | 4032,3024 | 7 | *49* |
| InternVL2-8B | 313,234 | 19 | 19 |
| | 787,444 | 19 | 19 |
| | 4032,3024 | 19 | 19 |
| InternVL2-26B | 313,234 | 1 | *10* |
| | 787,444 | 11 | *45* |
| | 4032,3024 | 1 | *10* |

Table 2: Comparison of the maximum number of images supported per request for various image resolutions across different models. Higher values are better; best values in each row are italicized.

**EPD Supports a Higher Number of Images per Request:** We compare the maximum number of images per request supported by the disaggregated (EPD) and aggregated (DistServe or vLLM) systems across three multimodal models: MiniCPMv, InternVL2-8B, and InternVL2-26B. The experiment was conducted at three different image resolutions, with a fixed batch size of 1 and with 80% of the available memory allocated for the KV cache. As shown in Table 2, EPD handles more images per request than DistServe, for example, at $4032\times3024$ resolution, *7× more* for InternVL2-26B and *10× more* for InternVL2-8B. For InternVL2-8B, the limit of 19 images per request is due to its maximum context length. Without this constraint, a larger number of images per request could be supported.

**EPD Supports Higher Batch Sizes:** We compare maximum supported batch sizes for E and P stages across dif-

ferent settings of image resolution and models in Table 3. The number of images per request is fixed to 10, and the KV cache is allocated to utilize 80% of the available free memory. As seen, EPD significantly outperforms DistServe for both E and P batch sizes. For example, with InternVL2-26B at $787\times444$ resolution, EPD supports a batch size of 22 for encoding vs. DistServe's 1, achieving *22× improvement*. Similarly, for MiniCPM-V 2.6 at $787\times444$, EPD achieves a batch size of 29 vs. DistServe's 2 for prefill, a *14.5× improvement*.

| Model | Image Reso. | #Patch | DistServe (E, P) | EPD E | EPD P |
|---|---|---|---|---|---|
| MiniCPMv 2.6 | 313,234 | 1 | 7 | *49* | *86* |
| | 787,444 | 3 | 2 | *16* | *29* |
| | 4032,3024 | 10 | OOM | *4* | *9* |
| InternVL2-8B | 313,234 | 13 | 2 | *15* | 2 |
| | 787,444 | 3 | 9 | *67* | *10* |
| | 4032,3024 | 13 | 2 | *15* | 2 |
| InternVL2-26B | 313,234 | 13 | OOM | *6* | *1* |
| | 787,444 | 3 | 1 | *22* | *4* |
| | 4032, 3024 | 13 | OOM | *6* | *1* |

Table 3: Comparison of the maximum supported batch sizes for E and P stages across different models and image resolutions. Higher values are better; italicized values indicate the best in each row. OOM denotes cases where the model ran out of memory.

### 4.4. Ablation Study

In this section, we analyze the impact of various components of the proposed system.

| | #I/R=2 | #I/R=4 | #I/R=6 | #I/R=8 |
|---|---|---|---|---|
| EPD | 0.92 | 1.02 | 1.14 | 1.74 |
| w/o IRP | 1.46 (1.6x) | 2.47 (2.4x) | 3.37 (2.9x) | 4.27 (2.5x) |

Table 4: Effect of ablating IRP feature from the proposed system on TTFT (s). Disabling IRP negatively affects the TTFT (up to *2.9x worse*) for various multiple images/ request (#I/R). Results are averaged over 100 requests.

**Effect of IRP:** We analyze the effect of ablating IRP, presented in Section 3.2.2. The experimental settings are the same as the TTFT experiment in Section 4.2. As shown in Table 4, removing the IRP feature negatively affects TTFT across various images/ request. Moreover, the degradation exacerbates as the number of images/ request increases. This is because the IRP feature allows parallelization of encoding load within the same request across multiple GPUs.

**Effect of Offline Optimizer:** By default, the EPD system collects workload samples and finds the optimal configuration offline using the optimizer. To demonstrate its effect, we conduct an experiment without using the optimizer and

|        | Goodput (r/s) ↑ | TTFT (s) ↓ | TPOT (s) ↓ |
|--------|-----------------|------------|------------|
| EDP    | 1.25            | 2.12       | 0.031      |
| w/o Opt. | 0.56 (2.2x)   | 4.48 (2.1x) | 0.025 (0.8x) |

Table 5: Ablating the offline optimizer reduces goodput by 2.2× on average when configurations are selected randomly. ↓ indicates lower is better; ↑ indicates higher is better.

select configurations randomly. Specifically, we uniformly sample 10 configurations at random and report the expected value of the performance metric in the second row of Table 5. For a fair comparison, when evaluating TTFT and TPOT, we maintain the same request rate of 1.25 r/s, which corresponds to the goodput of the EPD system (see Appendix E.4 for more details). As shown in Table 5, disabling the optimizer causes significant performance degradation in terms of both goodput and TTFT. This highlights the importance of the optimizer, especially when the EPD system has many tunable configurations.

**Effect of Dynamic Role Switching:** To analyze the impact of dynamic role switching, we conduct a controlled experiment simulating a shift in workload characteristics. Specifically, we generate 100 requests following the same static workload configuration as described in Section 3.2.4. However, to introduce an artificial workload change, the first 10 requests generate 50 output tokens, while the remaining 90 requests generate 500 output tokens. The request arrival rate is fixed at 3 requests per second. The results are shown in Table 6.

|        | Latency (s) ↓ | TTFT (s) ↓ | TPOT (s) ↓ |
|--------|---------------|------------|------------|
| EPD    | 28.01         | 1.42       | 0.05       |
| w/o Switch | 61.10 (2.2x) | 1.33 (0.9x) | 0.12 (2.4x) |

Table 6: Ablating dynamic role-switching from EPD degrades TPOT by 2.4× and increases end-to-end latency by 2.2×. Results are averaged over 100 requests with one 4K image each. ↓ indicates lower is better.

Without dynamic worker migration, the system performs poorly because it remains fixed in the initial configuration (5E1P2D, optimized offline for 50 tokens) and is unable to adapt to the increased decoding demand. In contrast, the EPD system with migration dynamically reconfigures itself (2E1P5D) to handle the new workload (500 output tokens) by shifting three E instances to D, resulting in approximately 2× better performance.

### 4.5. Adaptation to Neural Processing Units (NPUs)

We present the results of adapting the proposed EPD framework to Huawei Ascend NPUs. For more details on the implementation and experiments, see Appendix F.

First, we compare EPD-NPU against the vLLM and Dist-Serve baselines on the InternVL2-8B LMM. We evaluate SLO attainment, defined as TTFT ≤ 8.5s and TPOT ≤ 0.12s on eight 4K images per request. We used the same settings as described in Section 4.1, except that we employed a heavy encoding workload of eight $4032 \times 3024$ images per request. The optimal configuration for this workload was found to be 5E2P1D, and the corresponding results are presented in Figure 9. As shown, EPD is the only configuration that achieves the SLO requirements, while the other baselines fail to meet the SLOs entirely, even at low request rates.

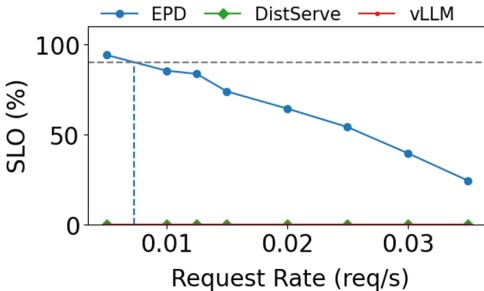

Figure 9: SLO attainment (↑) on NPUs under varying request rates for a synthetic workload using the InternVL2-8B model. EPD maintains positive SLO attainment under strict TTFT constraints, while baselines fail to meet the criteria.

Next, we compare the TTFT of EPD-NPU against vLLM. EPD-NPU achieves a 35.2% improvement over vLLM, surpassing the 24.4% improvement observed in the GPU scenario. This additional ∼10% improvement is attributed to the proportionally heavier encoding workload of LLMs on NPUs compared to GPUs. Please refer to Appendix F.1 for more details. As a result, disaggregating the encoding from the prefill phase yields greater benefits on NPUs than on GPUs.

## 5. Conclusion

In this paper, we proposed a novel approach for optimizing LMM systems via disaggregation of key processing stages. By separating the encoding, prefill, and decoding phases into distinct stages, our system provides greater flexibility in resource allocation, enabling more efficient management of computational and memory resources. This disaggregation, together with dynamic resource allocation, asynchronous token transfer, and advanced parallelization strategies, directly addresses several critical challenges in LMM deployment, including latency reduction, memory optimization, and efficient computational resource usage. We validated the effectiveness of our system through extensive experiments. Finally, we outline the system's limitations and future directions in Appendix B and Appendix C, respectively.

## Impact Statement

This paper presents work whose goal is to advance the field of Machine Learning. There are many potential societal consequences of our work, none which we feel must be specifically highlighted here.

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

# A. Additional Experiments

## A.1. Adaptation to Audio modality

In this section, we extend EPD to the audio modality to evaluate its effectiveness beyond vision tasks, highlighting the method's generalizability. Specifically, we evaluate EPD in an online, audio setting using the `ultravox-v0_3` model, based on LLaMA3.1-8B. To simulate an encode-intensive workload, each request is configured to include 24 audio files. Evaluation metrics include SLO attainment—defined as TTFT $\leq$ 2.0 s and TPOT $\leq$ 0.025 s—and goodput (requests per second, r/s). As shown in Table 7, EPD consistently outperforms vLLM and DistServe across all request rates, demonstrating superior efficiency and reliability. Combined with prior results in the video domain, these findings reinforce EPD's robustness and generalizability across modalities.

| Rate (r/s) | SLO Attainment Rates ($\uparrow$) | | |
| --- | --- | --- | --- |
| | vLLM | DistServe | EPD |
| 0.10 | 0.99 | 0.99 | *0.99* |
| 0.25 | 1.00 | 0.94 | *0.99* |
| 0.50 | 0.99 | 0.89 | *1.00* |
| 1.00 | 0.91 | 0.72 | *0.96* |
| 1.10 | 0.87 | 0.69 | *0.93* |
| 1.15 | 0.87 | 0.68 | *0.93* |
| Goodput (r/s) $\uparrow$ | 1.01 | 0.45 | *1.16* |

Table 7: SLO attainment results ($\uparrow$) for *online audio benchmarking* with `ultravox-v0_3` (24 audio files per request). All baselines use 4 GPUs: vLLM operates in data-parallel (DP) mode, DistServe uses a 3P1D configuration, and EPD adopts a 2E1P1D setup. EPD achieves consistently high SLO attainment and the highest goodput across all request rates.

## A.2. EPD Supports Larger KV Cache Sizes

In this section, we compare the maximum KV cache size that can be allocated across various baselines. The batch size is fixed at 1, and the image resolution is set to 4032×3024. As shown in Table 8, EPD supports significantly larger KV cache sizes than DistServe. For instance, for the InternVL2-26B model at 10 images per request, EPD can support a KV cache size of 80% vs 36% for DistServe, representing a *2.2×* *improvement*. Notably, in certain scenarios—such as with the InternVL2-26B model and 20 images per request—the DistServe system encounters an OOM error, indicating that it cannot process 20 images even with a KV cache size of 0. Additionally, some configurations encounter an Out of Context Limit (OOCL) error, where the large number of encoding tokens generated by high image counts exceeds the LLM's context limit during the prefill stage.

| Model | # Images/Req. | DistServe | EPD |
| --- | --- | --- | --- |
| MiniCPM-V 2.6 | 5 | 86% | *99%* |
| | 10 | 74% | *97%* |
| | 20 | 49% | *95%* |
| | 40 | OOM | *92%* |
| | 80 | OOM | OOCL |
| InternVL2-8B | 5 | 94% | *95%* |
| | 10 | 89% | *91 %* |
| | 20 | OOCL | OOCL |
| InternVL2-26B | 5 | 67% | *89%* |
| | 10 | 36% | *80%* |
| | 20 | OOM | *63%* |
| | 40 | OOM | OOCL |

Table 8: Comparison of maximum supported KV cache size (in terms of percentage of free memory) on prefill node for various #images/ request. Image resolution fixed to 4K. Higher (*italicized*) is better.

## A.3. Throughput Comparison for Offline and Heterogeneous scenario

In this section, we compare the throughput of the EPD and DistServe methods under both offline and heterogeneous settings. In the offline scenario, a batch of requests is submitted in advance, allowing the system to process them overnight with the goal of maximizing end-to-end (E2E) throughput. In the heterogeneous setting, we consider a cluster composed of GPUs with varying computational and memory capacities.

Our motivation is as follows: in the DistServe system, both the encoding and prefill stages are executed on the same worker, requiring the corresponding memory to be allocated on a single GPU. However, GPU memory is limited, and this constraint becomes particularly problematic in heterogeneous environments that include low-end GPUs. In such cases, the total memory demand of the encoding and prefill stages may approach or exceed the available memory, significantly restricting the batch size or even rendering the DistServe method infeasible on these devices. Thus, we consider a controlled scenario in our experiment where the memory usage of the prefill stage in DistServe remains within the capacity of low-end GPUs, but only permits a minimal batch size of 1 for both encoding and prefill stages.

We conduct this experiment using 8 A800 GPUs. For the EPD system, we use the default `5E2P1D` configuration, which allocates 5, 2, and 1 GPUs to the encoding, prefill, and decoding stages, respectively. The corresponding maximum batch sizes are set to 8 for encoding, 8 for prefill, and 128 for decoding. In the DistServe system, we adopt the `7P1D` setup, with maximum batch sizes of 1 and 128. Here, the encoding and prefill steps are conducted on the 7 P workers. The workload consists of 1,000 requests, each containing a single image, a simple prompt ("What is the

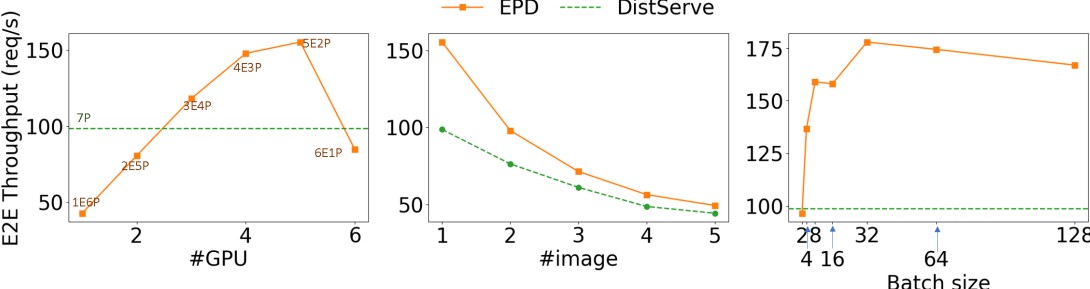

Figure 10: **Left:** Impact of varying the number of encoding workers in the EPD method. The notation `xEyP` denotes a configuration with `x` encoder and `y` prefill workers. The DistServe method uses a fixed `7P` configuration, assigning 7 workers to handle both encoding and prefill steps. **Middle:** Effect of the number of images per request on end-to-end throughput. **Right:** Sensitivity to encoding and prefill batch sizes, where these two numbers are set equal.

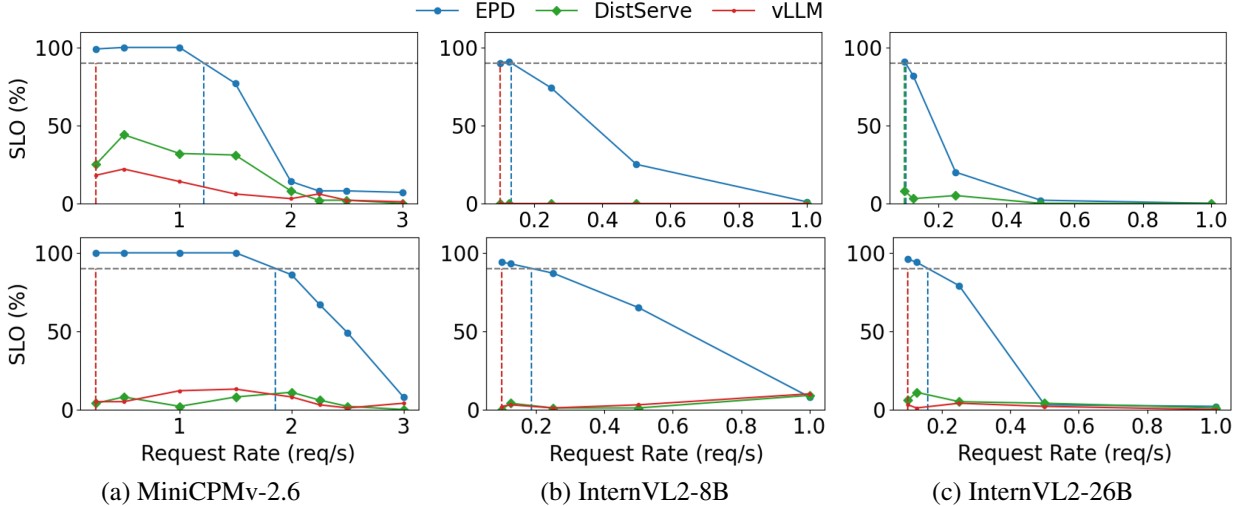

Figure 11: SLO attainment (↑) for end-to-end inference across multiple models and image counts per request. Subfigures (a), (b), and (c) correspond to MiniCPM-V 2.6, InternVL2-8B, and InternVL2-26B, respectively. The top and bottom rows show results *for 6 and 8 images per request*. EPD consistently outperforms all baselines, demonstrating robust performance as image count increases..

content of this image?"), and a maximum of 10 output tokens. We perform an ablation study by tuning the system's hyperparameters as described in Sec. 3.2.3, and analyze their impact on throughput. The results are presented in Fig. 10.

The left plot demonstrates the importance of selecting an appropriate GPU configuration. The algorithm detailed in Sec. 3.2.3 can automatically select the `5E2P` configuration as the optimal setup for maximizing end-to-end (E2E) throughput. The middle plot shows that EPD achieves higher throughput when the number of images per request is small, indicating that its disaggregated architecture effectively mitigates encoder-side compute bottlenecks. The right plot highlights that EPD is relatively insensitive to encoding and decoding batch sizes. This enables users to

either manually choose a large batch size or rely on the algorithm in Sec. 3.2.3 to automatically determine an optimal configuration.

### A.4. SLO Attainment for Higher Number of Images

In this section, we show additional results from the experiment in Section 4.1 pertaining to 6 and 8 images per request. The results are presented in Figure 11. As seen, the SLO attainment for EPD begins to decline as number of images rise, particularly at higher request rates. This decline reflects the increasing computational demand associated with processing multiple high-resolution images within a single request. Nevertheless, EPD continues to outperform all baselines, which struggle even at low request rates and fail to scale effectively under increased workloads.

## B. Limitations

While disaggregated serving offers significant benefits in meeting strict service-level objectives, it is not universally optimal. In scenarios where throughput is the primary concern and SLOs can be relaxed, traditional aggregated serving can outperform disaggregated approaches due to superior GPU utilization. This is largely attributed to the absence of inter-stage communication overhead and pipeline inefficiencies/ bubbles that can arise from resource imbalance in disaggregated setups.

From a cost-efficiency perspective, disaggregated serving is most beneficial when latency SLOs are non-negotiable. In contrast, when maximizing throughput per dollar is the primary goal and latency constraints are minimal, aggregated configurations are often more cost-effective. This can be attributed to the reduced communication cost and orchestration overhead that result in higher GPU utilization.

Specific to the EPD disaggregation strategy, the encoding and prefill stages are predominantly compute-bound. As a result, the primary performance gains from disaggregation stem from stage-specific finetuning of batching, scheduling, and parallelization strategies, especially IRP for parallelizing encoding load of a request across multiple GPUs. However, these gains are sensitive to pipeline inefficiencies, for instance pipeline bubbles can erode the latency benefits. Thus, careful tuning of resource allocation and responsive worker migration is required to ensure these stages are well-balanced for a given model and workload.

## C. Future Work

Although the encode and prefill stages have similar compute characteristics, their memory usage profiles differ substantially. This asymmetry presents a promising opportunity to explore heterogeneous hardware configurations. For instance, using high-memory high-compute GPUs for prefill and low-memory but high-compute units for encoding could further enhance efficiency and cost-effectiveness in disaggregated setups.

Another area of exploration is reducing the overhead of visual token migration, which can be a bottleneck in multimodal disaggregated pipelines. Compression techniques, particularly those leveraging the inherent redundancy in visual tokens as identified in token pruning literature (Alvar et al., 2025), may significantly reduce transfer latency and cost.

Lastly, disaggregated serving introduces new possibilities in privacy-aware inference (Liu et al., 2020; Huang et al., 2020; Singh et al., 2022), especially in edge-cloud environments. For example, encoding could be performed on edge devices, keeping raw, privacy-sensitive images local. Only the encoded representations would be transmitted to the cloud for prefill and decode stages, minimizing privacy risks without compromising model capability. This hybrid architecture could form the foundation for privacy-preserving multimodal systems.

## D. Optimizer Details

We recapitulate the configuration optimization problem:

$$\max_{(\mathbf{p},\mathbf{b},\mathbf{s})\in\mathcal{X}} f(\mathbf{p}, \mathbf{b}, \mathbf{s}) - \beta \cdot \text{cost}(\mathbf{p}) \qquad (2)$$

For the search space $\mathcal{X}$, there could be implicit constraints during the configuration search. For example, the total number of GPUs must not exceed the available resources (e.g., 8 or 16). Alternatively, if the cloud service provider intends to fully utilize all available 8 GPUs, an implicit constraint could enforce the number of GPUs to be exactly 8. These constraints serve to reduce the search space. The system configuration involves parallelization configurations $\mathbf{p}$, maximum batch size configurations $\mathbf{b}$, and scheduling configurations $\mathbf{s}$. These variables are vectors, where each element corresponds to the configuration of an individual instance. Each instance is capable of handling requests independently, including managing (sub-)workers for tensor parallelism and pipeline parallelism. Instances within a stage process different requests in parallel, a concept referred to as data parallelism. Note that $\mathbf{p}$, $\mathbf{b}$, and $\mathbf{s}$ can have variable lengths, as the number of instances is itself a configurable parameter. For the $i$-th instance, we denote its stage as $\text{Stage}_i \in \{E, P, D\}$, where $E$ represents the encoder stage, $P$ represents the processing stage, and $D$ represents the decoder stage.

- **Parallelization**: Let $\mathbf{p}$ denote the vector of parallel configs for all instances. For the $i$-th instance, if it is a prefill or decoding instance, then its config $p_i$ includes: $p_i^{\text{TP}}$, the number of GPUs used for tensor parallelism; and $p_i^{\text{PP}}$, the number of GPUs used for pipeline parallelism. If it is an encoding instance, considering IRP does not require communication, which is better than TP, we only use IRP. Therefore for encode, we overload the symbol $p_i^{\text{TP}} = p_i^{\text{IRP}}$ to denote the number of GPUs used for IRP. If the cost per GPU is a constant $c$, then the total cost is: $cost(\mathbf{p}) = c \sum_{p_i \in \mathbf{p}} (p_i^{\text{TP}} \times p_i^{\text{PP}})$.

- **Max Batch Size** determines how many requests are processed simultaneously during the encoding, prefill, and decoding stages. Let $\mathbf{b}$ denote the batch size config for all instances. For $i$-th instance, $b_i$ is its max batch size. This config involves a trade-off between latency and throughput. Larger batch sizes improve throughput by enabling parallel processing but may increase latency if the stages become compute-bound.

- **Scheduling** involves two main decisions: First, which workers should each request be assigned to? Second, how should the order of requests be determined within a

worker queue? To solve these decisions, we adapt strategies from the DistServe framework (Zhong et al., 2024). In the encoding stage, when a request arrives, it is assigned and pushed to an instance queue. Between different stages, global queues are used, and each available engine pulls proactively from the queues. Possible assignment strategies in the encoding stage include Round-Robin or Least-Loaded First for assigning requests. Once a request is assigned to a worker's queue, we can apply ordering strategies like first-come-first-serve or shortest-job-first, or more complex strategies that prioritize requests based on their Service Level Objectives (SLOs). For simplicity, we constrain that all instances within the same stage share the same scheduling strategy.

# E. Implementation Details

EPD is a fully capable distributed serving system for LMMs, comprising several key components: a load estimation module, a resource allocation module, a RESTful API frontend, and a multimodal-aware orchestration layer. The entire framework is implemented with a mix of Python and C++/ CUDA implementations, ensuring superior scalability and performance. To facilitate integration, we repurpose the distributed execution engine from vLLM, which supports numerous popular LLMs and LMMs, allowing easy adaptation of new models into our disaggregated framework with minimal effort.

The API interface adheres to OpenAI's multimodal specifications, enabling users to specify parameters such as output length, temperature, and multimodal data inputs.

The scheduler is specifically designed for the disaggregated EPD framework, dynamically managing batch sizes and enabling asynchronous execution of the encoding, prefill, and decoding phases. The load estimation module ensures efficient GPU allocation across these phases, adapting to changing workload demands in real time.

Our repurposed distributed execution engine uses Ray actors to implement GPU workers, which manage multimodal and key-value caches, and coordinate the independent execution of the encoding, prefill, and decoding tasks. Furthermore, it supports 3D parallelism, incorporating Data Parallelism (DP), Tensor Parallelism (TP), and Pipeline Parallelism (PP) to maximize resource utilization and scalability.

The orchestration layer includes custom CUDA kernels optimized for parallelism in the encoding and prefill phases. These kernels enable efficient management of paged multimodal caches and ensure seamless asynchronous transfer of multimodal tokens between encoding and prefill GPUs. The orchestration layer oversees the execution of encoding, prefill, and decoding instances, handling tasks such as request distribution, KV cache transmission, and result

aggregation. For efficient data movement, the system employs NCCL for inter-node GPU communication and asynchronous `CudaMemcpy` for intra-node transfers, ensuring smooth operations without disrupting GPU computations across the disaggregated EPD framework.

## E.1. Hyper-parameters in EPD system

We conducted our experiments using a cluster of 8 NVIDIA A100 GPUs (82GB). Each server was equipped with 128 CPUs and 1TB of RAM. The CUDA version was 12.2. Flash attention-2 was used for the attention implementation. We use FP16 precision for all experiments.

To ensure a fair comparison, we standardized key performance-affecting settings of the inference engine across all baselines. Specifically, these include a block size of 16; a maximum of 2048 blocks per request; context tokens capped at 49,152, and decoding tokens at 81,920 per batch. The scheduling policy for all stages was set to First-Come-First-Served (FCFS). Further, to allow enough resources for memory-heavy multimodal requests to execute, KV cache GPU utilization was set to 50%, and the maximum number of multimedia data of 32 was imposed per prompt. The size of the multimodal cache was fixed to 3000 across all models, and the vLLM inference engine was run in eager mode. Finally, for the vLLM inference engine, we used the version 0.6.1.post1 which represents a stable version for multimodal inference.

In our online experiments, requests were sent to the inference engine using a Poisson arrival process with a fixed $\lambda$, representing the number of requests per second. Each trial was executed until 100 requests were completed, ensuring sufficient data for consistent performance analysis. Further, to optimize TTFT and TPOT in this latency-sensitive setting, we disabled batching by setting batch sizes to 1 for the encoding, prefill, and decoding stages.

## E.2. LMM Details

MiniCPM-V 2.6 integrates a SigLip-400M vision encoder, comprising 400 million parameters, with a Qwen2-7B language model, containing 7.6 billion parameters, culminating in 8 billion parameters. It excels in tasks involving single-image, multi-image, and video comprehension, even surpassing GPT-4V in these domains (Yao et al., 2024).

InternVL2-8B features an InternViT-300M-448px vision encoder with 300 million parameters, paired with an internlm2_5-7b-chat language model comprising 7.7 billion parameters, totaling 8 billion parameters. Similarly, InternVL2-26B combines an InternViT-6B-448px-V1-5 vision encoder with 6 billion parameters and an internlm2-chat-20b language model containing 20 billion parameters, resulting in 26 billion parameters.

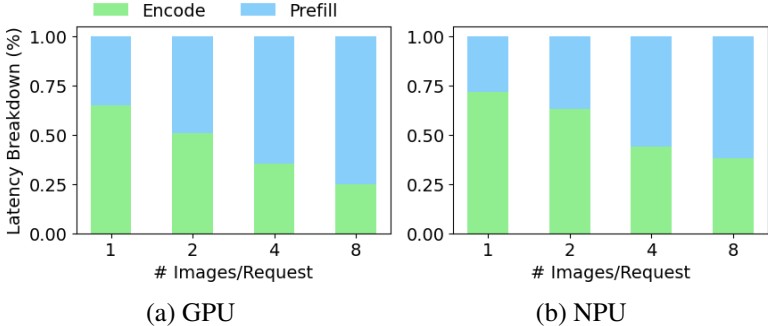

Figure 12: Breakdown of latency for encode and prefill stages using the InternVL2-8B model across varying numbers of images per request. Subfigures (a) and (b) show results on GPU and NPU, respectively. Light green denotes encode latency and light blue indicates prefill latency. NPUs demonstrate distinct latency characteristics compared to GPUs as input size increases.

### E.3. SLO Criteria

Table 9 outlines the SLO criteria for TTFT and TPOT across various models and different numbers of images per request (#I/R). These criteria are empirically derived based on the characteristics of the underlying models, such as the computational complexity of the MME and LLM. We also consider what is realistically achievable by both our method and the baselines on a fixed number (8 GPUs) used in experiments. Across models, as #I/R increases, there is an approximately linear increase in the TTFT criteria due to the higher encoding load and additional tokens generated during the prefill stage. In contrast, TPOT requires only minor adjustments since it is not directly impacted by changes in #I/R.

| #I/R | MiniCPM-V 2.6 | | InternVL 8B | | InternVL 26B | |
|------|------|------|------|------|------|------|
| | TTFT | TPOT | TTFT | TPOT | TTFT | TPOT |
| 2 | 1.40 | 0.04 | 1.20 | 0.05 | 3.50 | 0.07 |
| 4 | 2.60 | 0.04 | 2.40 | 0.06 | 7.05 | 0.08 |
| 6 | 3.90 | 0.06 | 3.55 | 0.09 | 11.00 | 0.95 |
| 8 | 5.10 | 0.06 | 5.00 | 0.18 | 15.00 | 0.15 |

Table 9: TTFT and TPOT values (in seconds) used as SLO thresholds for different models and image counts per request (#I/R). Respective values are shown for MiniCPM-V 2.6, InternVL2-8B, and InternVL2-26B models.

### E.4. Details for the Ablation Experiment of Offline Optimizer

In this experiment, 100 User requests arrive in real-time with each request containing 6 images, and they all request for the MiniCPMv model. For simplicity, we limit the configuration space to a restricted search space, where each data-parallel (DP) worker uses the same batch size configuration, and both TP and PP are fixed to 1. We explore the batch size configurations for workers across different stages, the number of DP workers in each stage, and the decision to enable IRP. The config identified by the optimizer is as

follows: batch sizes for the E, P, and D stages are 2, 1, and 128, respectively; the number of workers in each stage is 6, 1, and 1; and IRP is enabled. We uniformly random-sample 10 configs from the search space. To maintain the same computational cost, all 8 GPUs are utilized across different configs. Consequently, the search space is constrained to leverage 8 GPUs, which can be enforced through rejection sampling. The sample mean of the performance metric (goodput, TTFT, and TPOT) is presented in Table 5. Recall that the goodput metric is defined as the maximum request rate while maintaining an SLO attainment of no less than 90%. For a fair comparison, When evaluating the TTFT and TPOT, we maintain the same request rate of 1.25 r/s, which is same as the goodput of the EPD system.

## F. Extending EPD to NPU

We extend our EPD framework to a cluster of Neural Processing Units (NPUs) for two purposes: (1) to investigate the impact of disaggregating (E) from (P) on NPUs which tend to have higher encode-to-decode latency ratio and (2) to demonstrate the generalizability of EPD.

### F.1. NPUs having higher encode-to-prefill latency ratios

When investigating architectures and potential benefits of an EPD-NPU framework, we first profiled the LMMs tested in this paper for their encoding and prefilling latencies to understand the impact of the characteristics described above. We calculated the average encode and prefill stage latencies of 10 requests. Across the different LMMs, NPUs spent more time on encode than prefill across these models when compared to GPUs. Figure 12 shows the latency breakdown between encode and prefill for InternVL2-8B across a range of #I/R. Across the models and different #I/R, we found a ∼10-20% larger encode-to-prefill latency ratios in NPU than in GPU. This trend was consistent across different large multimodal models (LMMs), indicating that NPUs tend to

spend a greater proportion of time in the encoding phase compared to GPUs.

Since the encode in NPU is a larger portion of the TTFT when compared to GPU, we hypothesized that we would see more benefit from EPD and IRP towards NPU. Thus, we were motivated to adapt EPD for NPU.

### F.2. Implementation Details for NPU

The EPD-NPU system was implemented in a similar manner to our EPD-GPU system as described in Appendix E. All EPD-NPU experiments utilize IRP. The EPD-NPU distributed serving system was developed and deployed on a cluster of 910B3 NPUs, each with 64GB of high bandwidth memory. There are two major differences between the NPU and GPU implementations:

- **Ascend-vLLM**: EPD-NPU leverages Ascend-vLLM rather than vLLM which is an adaptation of vLLM on Ascend NPU hardware that utilizes the Compute Architecture for Neural Networks (CANN) software stack rather than CUDA.
- **Container-based deployment**: Each encode, prefill, and decode instance is deployed separately as an API for the scheduler engine to call. These instances run in data parallel mode. The scheduler engine simply sends a POST request to the corresponding API with some minimal

request-specific information (e.g., request ID, sampling parameters, block IDs to pull) and the remaining implementation mirrors that of EPD-GPU. This container-based deployment allows our framework to be easily scalable and portable to the cloud, providing intriguing future works regarding efficient utilization and scheduling of resources for users.

### F.3. Experimental Settings for NPU

For EPD-NPU, We use eight 910B3 NPUs and evaluate the system by running the controlled experimental setup (as described in the experiments section of the main paper) in an online setting. The number of input tokens was 22, length of output tokens was 10 with the end-of-sequence token ignored. The SLO requirements (8.5 and 0.12 for TTFT and TPOT, respectively) were also selected in a similar manner as the GPU experiments. The maximum model length was set to 27000, and KV Cache utilization was set to 83%. EPD-NPU also used a version of Ascend-vLLM (version 0.6.3.post1) with CANN version 7.6. Unless otherwise stated, the hyper-parameters are identical to those described in E.1.

We evaluate EPD-NPU using InternVL2-8B LMM against two baselines: (1) Ascend-vLLM and (2) Ascend-vLLM PD disaggregation for the heavy workload of 8 images (4032x3024 resolution) per request.

