# OpenReview forum: "Efficiently Serving Large Multimodal Models Using EPD Disaggregation"
_ICML.cc/2025/Conference — ICML 2025 poster_

### Official Review · Reviewer_rZNR · 2025-02-17

**Overall Recommendation:** 4

**Summary:**

--- score updated from 3 (weak accept) to 4 (accept) after the rebuttal. ---

----

Recently Disaggregated Inference was proposed to use separate nodes for prefill and decoding whilst serving LLMs. This allows to more easily control SLO times like inter token latency or time to first token.

The paper generalizes this setting form LLMs to Multimodal models, proposing to additionally disaggregate the Encoding phase of images or videos. This idea is very relevant and timely, but also conceptually straightforward. The paper conducts many studies ablating the different aspects of the solution and convincingly making a case for the proposed solution.

Their key innovations are a/ an asynchronous token transfer between the workers, b/ Intra-request parallelism which allows to further speedup encoding by running independent image encodings of a single request across workers c/ an alogrithm to allocate the resources between encoding/prefill/decode and d/ an approach to dynamically switch the roles.

**Claims And Evidence:**

Yes the empirical claims are supported by experimental results.

**Essential References Not Discussed:**

n/A

**Experimental Designs Or Analyses:**

The experiments seem mostly adequate, see also above. However, the code is not provided. In particular for the optimized resource allocation it is left unclear how the algorithm works.

**Methods And Evaluation Criteria:**

Yes the Evaluation mostly makes sense.

I would propose to also look at cost per served request. One criticism of Disaggregation is that it is not the most cost-efficient solution. This is fine, the authors are also not necessarily claiming this, but it could be clearly discussed as a limitation with also data illustrating when this happens.

**Other Comments Or Suggestions:**

N/A

**Other Strengths And Weaknesses:**

The major weakness is that no code is provided. Especially for the Optimized resource allocation and the dynamic role switching this is crucial IMO. There are not enough details provided to even reproduce the results of the paper. Since this paper mainly proposes an  ellaborate efficient software for a rather simple scientific problem, I consider code and reproducibilty crucial for this work.

**Questions For Authors:**

Can you please include the code in your rebuttal? Then I will support acceptance.

**Relation To Broader Scientific Literature:**

Prior work is appropriately discussed. The current submission fits timely into this evolving field

**Theoretical Claims:**

There are no theoretical claims

---

> ### Author Rebuttal · Authors · 2025-04-01
>
> We thank the reviewer for the thoughtful comments, recognition of the importance and timeliness of our work, and the positive assessment of our empirical validation. Below we address the main concerns raised:
>
> ---
>
> ### **Q1: Cost per served request and disaggregation tradeoffs**
>
> > _"I would propose to also look at cost per served request. One criticism of Disaggregation is that it is not the most cost-efficient solution... it could be clearly discussed as a limitation with also data illustrating when this happens."_
>
> **Response:**
> We appreciate this insightful comment and fully agree that **disaggregation is not always the most cost-efficient approach**, particularly in scenarios where **tight Service-Level Objectives (SLOs) are not required** and **compute resources (e.g., GPUs) are limited**. For instance, in **offline batch processing**, where throughput is the sole objective, aggregated architectures can achieve **higher GPU utilization** and benefit from **lower inter-stage communication overhead**, making them more cost-effective.
>
> We will **explicitly discuss this limitation in the revised version** and provide a more nuanced view of when disaggregation may or may not be cost-efficient.
>
> That said, recent work such as [1] demonstrates that **disaggregation can sometimes yield even better throughput**, especially when **compute requirements of stages is significantly different**. By optimizing batching, parallelization and scheduling strateges, systems like DeepSeek-V3 (which disaggregates prefill and decode) are able to achieve better throughput. Our **Throughput in Offline Settings** analysis in the Supplementary Material supports this finding, showing that disaggregation can reduce computation bubbles and improve utilization when stages are well balanced.
>
> However, in **cloud-scale, interactive serving systems**, where **individual SLO guarantees (e.g., time-to-first-token, time-per-output-token)** become crucial for user experience, disaggregation is critical for cost efficiency.
>
> In such cases, **ensuring SLOs with fully aggregated systems requires significant over-provisioning**, which can lead to **higher overall cost** than disaggregated setups. Disaggregation allows each stage to be optimized independently, reducing interference and tradeoffs between stages, that result in better control of per-request level statistics like TTFT and TPOT.
>
> This is evident from experiments in Section 4.1. EPD has better SLO attainment than aggregated setups for same number of GPUs. Hence, for aggregated setups to match the SLOs of EPD, they need to overprovision, resulting in higher overall cost.
>
> [1] DeepSeek-AI. DeepSeek-V3 Technical Report. arXiv 2412.19437, 2025
>
>
> ---
>
>
> ### **Q2 & Q3: Lack of code release, especially for resource allocation and role switching**
>
> > _"The major weakness is that no code is provided.  Especially for the Optimized resource allocation and the dynamic role switching.. I consider code and reproducibilty crucial for this work."_
>
> **Response:**
> We understand and agree with the reviewer that **code and reproducibility are vital**, especially for software systems that propose architectural and scheduling innovations.
>
> We had planned to release the code as part of the camera-ready version (post-acceptance), but in light of the reviewer’s concern, we have **made the code available during the review period** at the following anonymous link:
>
> > **Anonymous Code Repository:** [link](https://drive.google.com/drive/folders/1cEyGCPw54EkgBjZs73m-SZq51M2etD0X) (released under Apache license)
>
> This release includes:
> - The **full implementation of EPD**, including stage clusters, worker orchestration, cache management, and bridges
> - Our **custom resource allocator** with multiple backends (runtime, simulation)
> - The **dynamic role switching mechanism**, including metrics collection and migration orchestration
> - Scripts for **reproducing main experimental results**
> - **Readme file** located at <project_root>/README.md
>
> We hope this supports the reviewer’s confidence in the rigor and reproducibility of our work.
>
> ---

---

### Official Review · Reviewer_QF6q · 2025-03-14

**Overall Recommendation:** 3

**Summary:**

To address the negative impact of the multimodal encoding stage on key Service Level Objectives (SLOs), this paper proposes the Encode-Prefill-Decode (EPD) Disaggregation framework, which allocates the encoding, prefill, and decode stages to independent computing resources. Specifically, this work introduces: (1)A new mechanism for caching multimedia tokens to improve transmission efficiency. (2)A novel method for parallelizing encoding loads within the same request. (3)A resource allocation module optimized for disaggregated inference. (4)A dynamic role-switching approach to adapt to changing workload characteristics.
Experiments on MiniCPMv-2.6, InternVL2-8B, and InternVL2-26B demonstrate that, compared to non-disaggregated methods, the proposed framework significantly improves memory efficiency, batch processing capacity, image handling capability, KV cache utilization, TTFT, and E2ETP. Overall, this paper presents a clear motivation, an intuitive and well-founded approach, and compelling experimental results, demonstrating strong practical applicability.

**Claims And Evidence:**

This paper claims three key contributions: Intra-Request Parallelization (IPR), Resource Allocation Optimization Modeling, and Dynamic Role Switching. After reading the full paper, I think all claims are clear and well-supported by evidence.

**Essential References Not Discussed:**

N/A

**Experimental Designs Or Analyses:**

The experiments in this paper primarily focus on the following four aspects: (1) End-to-end SLO evaluation (EPD significantly outperforms the baselines) (2)Computational resource utilization efficiency (comparison of maximum supported images and batch sizes) (3)Ablation studies on key modules (IPR, DRS, and optimizer) (4)Analysis of TTFT and TPOT (key inference performance metrics comparison). However, some in-depth analyses are missing in this paper. For example, why does TPOT perform better in the w/o Opt setting compared to w Opt in Table 4? Similarly, why is TTFT lower in the w/o Switch setting in Table 5? I hope the authors can provide reasonable explanations for these phenomena. Additionally, the paper only conducts experiments on the NextQA dataset. How does the proposed method perform on other more popular datasets, such as Video-MME? More comprehensive evaluations would strengthen the validity of the results.

**Methods And Evaluation Criteria:**

The proposed method is reasonable and its effectiveness is validated using mainstream models and datasets.

**Other Comments Or Suggestions:**

N/A

**Other Strengths And Weaknesses:**

N/A

**Questions For Authors:**

See above.

**Relation To Broader Scientific Literature:**

N/A

**Theoretical Claims:**

This paper primarily focuses on engineering-oriented inference optimization and lacks theoretical foundations. However, I believe this does not diminish its value.

---

> ### Author Rebuttal · Authors · 2025-04-01
>
> Thanks you! Glad to hear that the **motivation, design, and experimental results were found to be compelling**.
>
> Below we address the specific questions raised.
>
> ---
>
> ### **Q1: Why does TPOT perform better in the w/o Opt setting compared to w Opt in Table 4?**
>
> Thanks for pointing it out, here is the clarification. The optimization strategy may sometimes trade off TPOT to improve the global objective -- goodput.  In the experiment from Table 4, for EPD with an optimizer, the optimizer's objective is solely to maximize the goodput metric. Therefore, the small decrease in average TPOT is due to the optimizer automatically finding a better tradeoff between average TTFT and TPOT, given the workload statistics of encoding, prefill, and decoding.
>
> Specifically, the optimizer sacrifices average TPOT only slightly (from 0.025s to 0.031s) while significantly reducing average TTFT (from 4.48s to 2.12s). The final TTFT value of 2.12 s falls well below the 3.9 s requirement (recall that our experiment setup requires TTFT and TPOT to be 3.9s and 0.06s, respectively, as shown in Table 6).
>
> ---
>
> ### **Q2: Why is TTFT lower in the w/o Switch setting in Table 5?**
>
> Thank you for this observation, we’re happy to clarify. The **lower TTFT in the w/o Switch setting** is a direct consequence of its **static resource allocation**: the system remains locked in the initial **5E1P2D** configuration, which was optimized offline for short outputs (50 tokens). This setup overprovisions the **encoding stage**, resulting in a slightly lower TTFT of **1.33s** vs **1.42s** for EPD.
>
> However, this comes at a significant cost. The static configuration fails to adapt when the workload shifts to **longer output requests (500 tokens)**, and decoding becomes the primary bottleneck. As a result, the **TPOT deteriorates sharply to 0.12s**, more than **2.4× worse** than EPD (**0.05s**).
>
> By contrast, the **full EPD system**, equipped with **dynamic role switching**, detects this workload shift and **migrates 3 encoding workers to decoding**, transitioning to a **2E1P5D** configuration. While this reduces encoding capacity and leads to a slightly higher TTFT (**1.42s**), it **significantly improves decoding throughput**, yielding a much lower TPOT and overall faster response.
>
> Ultimately, this leads to a **2.2× improvement in average end-to-end latency**: **28.01s for EPD vs. 61.10s for w/o Switch**. This tradeoff highlights the strength of our dynamic reconfiguration strategy: it **prioritizes end-to-end latency** by resolving the true bottleneck in real time, even if that means marginally a particular stage.
>
> ---
>
>
> ### **Q3: How does the method perform on other datasets like Video-MME?**
>
> Thank you for the thoughtful suggestion. To strengthen the paper, we conducted additional experiments on the **Video-MME benchmark**, which complements NextQA (open-ended questions) by focusing on multi-choice video QA with diverse video lengths.
>
> We evaluated **SLO attainment** (TTFT ≤ 3.1s, TPOT ≤ 0.025s) on 100 randomly sampled examples using **MiniCPM-v2.6**, with 64 uniformly sampled frames per video, adhering to the MiniCPM frame setting reported on the Video-MME leaderboard. Results are shown below:
>
> **Table 1: SLO Attainment Rate (%) ↑ under Different Request Rates**
>
> | Method \ Rate (req/s)  | 0.5  | 1.0  | 1.5  | 2.0  | 2.25 | 2.5  | 3.0  |
> |------------------------|------|------|------|------|------|------|------|
> | **vLLM**              | 70   | 52  | 46   | 29   | 32   | 21   | 3   |
> | **DistServe**         | 72   | 57   | 31   | 28   | 24   | 19   | 0   |
> | **EPD (ours)**        | **99**  | **100**   | **99**  | **87**   | **65**  | **43**   | **9**  |
>
> **EPD consistently outperforms vLLM and DistServe** across all rates, demonstrating strong generalization to temporal multimodal workloads. This highlights the effectiveness of our proposed EPD beyond the NextQA dataset for temporal multimodal inputs like video.
>
> We also evaluated **TTFT latency** under varying video lengths (8–64 frames), at 1 req/s:
>
> **Table 2: TTFT (s) ↓ under Different Frame Counts**
>
> | Method \ # Frames/Video | 8  | 16  | 32  | 64  |
> |------------------------|------|------|------|------|
> | **vLLM**              | 0.42   | 0.82  | 1.59  | 3.11   |
> | **DistServe**         | 0.42   | 0.81   | 1.54   | 3.08   |
> | **EPD (ours)**        | **0.24**  | **0.30**  | **0.49**  | **1.00**   |
>
> EPD **consistently achieves significantly lower TTFT latency** than both vLLM and DistServe across all frame counts. Moreover, the **performance gap widens with increasing video length**—for instance, at 8 frames, EPD reduces latency by **42.9%**, and at 64 frames, the reduction reaches **67.5%** compared to DistServe. These results highlight EPD’s **superior scalability and robustness** under increasingly demanding video processing workloads.
>
> ---
>
> Additionally, feel free to explore our released code at this [link](https://drive.google.com/drive/folders/1cEyGCPw54EkgBjZs73m-SZq51M2etD0X).
>
> ---

---

### Official Review · Reviewer_afT5 · 2025-03-19

**Overall Recommendation:** 3

**Summary:**

After rebuttal: Thank the authors for the detailed comments. I'm keeping my recommendation of 3.

----------------------------------
The authors propose a novel Encode-Prefill-Decode (EPD) disaggregation framework for Large Multimodal Model (LMM) inference. The proposed approach decouples encoding and prefill stages, so that GPU resources can be allocated more efficiently based on the jobs' encode, prefill, and decode stages. The EPD framework includes intra-request parallelization (IPR) to shard a request into independent sub-requests, an optimizer to handle resource allocation, and a dynamic role switching mechanism to monitor the bottlenecks and switch the role of a GPU if needed. Evaluations are conducted on MiniCPM-V 2.6, InternVL2-8B, and InternVL2-26B using synthetic workloads and NextQA benchmark dataset on a cluster of 8x NVIDIA A100 GPUs, and achieved 71% reduction in time to first token (TTFT) and 57% reduction in end-to-end throughput (E2ETP).

**Claims And Evidence:**

Overall I find the claims clear and convicing. The authors proprose the EPD framework to address the LMM inference efficiency issue brought by the additional encoding stage to process raw multimodal inputs (compared to LMMs), which could require substantial GPU resources to encode them into tokens. The experiments show that the proposed EPD framework can improve multiple memory and latency metrics. Ablation studies are also conducted to show the effectiveness of each component.

I am not a big fan of the term Large Multimodal Model being used in the paper (I understand that InternVL calls themselves LLM). I think Vision Language Models would be more appropriate, as the advantage of the proposed EPD framework is largely based on the efficiency of the image encoding stage enabled by parallelism. We don't know if the same efficiency can be achieved for other modalities, e.g., audio.

**Essential References Not Discussed:**

N/A.

**Experimental Designs Or Analyses:**

The authors mainly compare the proposed approach with DistServe [Zhong et al., 2024], which implements the prefill decode (PD) disaggregation framework for LLMs (so no/little encoding compared to this paper). Overall the experimental designs are sound to me. In multiple experiments the authors use a resolution of 4032 x 3024, which arguably favors the proposed EPD approach due to the encoding cost.

**Methods And Evaluation Criteria:**

The authors use both synthetic datasets and the NextQA benchmark dataset for evaluation, which make sense to me. Metrics such as the supported number of images per request, the batch size, TTFT, TPOT, SLO attainment rate, and goodput are used to evaluate the proposed EPD framework, which are relevant to the claim being made.

**Other Comments Or Suggestions:**

N/A.

**Other Strengths And Weaknesses:**

The contributions of the paper are decent but not surprising. To be fair I feel VLM would be more appropriate in the title compared to LMM, as the advantage of the proposed EPD framework is largely based on the efficiency of the image encoding stage. We don't know if the same efficiency can be achieved for other modalities, e.g., audio.

**Questions For Authors:**

- Regarding Eq. (2), I think this falls into formalism and I cannot get much insight from it. I wonder if the authors can provide a concrete example. How should one use the formula in practice? How would the authors define the search space in their experiments?

- Have the authors considered other modalities such as audio in the evaluation? If not, do you think the proposed EPD framework can be generalized to other modalities?

**Relation To Broader Scientific Literature:**

I'm not an expert in this area. The paper can feel like a natural extension of the DistServe framework (plus Intra-Request Parallelism), which is also a disaggregation framework for LMMs, except that the proposed EPD framework takes into account of the additional the encoding stage. To me the results are sound and decent but not surprising. A lot of these are based on the fact that image and video inputs can be patchified and parallelized efficiently, and I wonder if this still holds for more sequential type of modalities.

**Theoretical Claims:**

N/A. This is an application paper.

---

> ### Author Rebuttal · Authors · 2025-04-01
>
> We thank the reviewer for the thoughtful and constructive feedback. Below, we address the specific concerns.
>
> ---
>
> ### **Q1: Use of the Term “LMM” and Generalization Beyond Vision-Language Models**
>
> We deliberately use the term **LMMs** instead of “VLMs” to reflect the **general applicability of EPD across modalities**, not just vision.
> EPD is designed for any model that consists of a modality-specific encoder followed by an LLM decoder. This includes **audio**, **video**, and other input types where encoders operate independently on local patches or chunks.
>
> Even for sequential modalities like audio or video, temporal structure is typically **reconstructed by the LLM using positional embeddings (e.g., RoPE)**—not by the encoder—making **intra-request parallelism and stage disaggregation effective across modalities**.
>
> To support this, we have extended our experiments to the **audio domain**, using the `ultravox-v0_3` model (`LLaMA3.1-8B` backbone) in an **online, encode-heavy setting** with **24 audio files per request**. Table 1 shows that EPD **outperforms both vLLM and DistServe across request rates**, achieving **higher SLO attainment** and **higher goodput**.
>
> **Table 1. Online Audio Benchmarking: SLO Attainment Rate (↑)**
> *SLO targets: TTFT ≤ 2.0s, TPOT ≤ 0.025s*
>
> | Method (Config) \ Rate (req/s) | 0.1 | 0.25  | 0.5  | 1  | 1.1  | 1.15 | Goodput (r/s) |
> |------------------------|------|------|------|------|------|------|------|
> | **vLLM**  (4D)        | 0.99   | 1.00   | 0.99   | 0.91   | 0.87   | 0.87   |   1.01  |
> | **DistServe**  (3P1D)   | 0.99   | 0.94   | 0.89   | 0.72   | 0.69   | 0.68   |  0.45  |
> | **EPD (ours)**  (2E1P1D)    | **0.99**   | **0.99**   | **1.00**   | **0.96**   | **0.93**   | **0.93**   |  **1.16**  |
>
> These results, along with our video experiments (see response to Reviewer QF6q), confirm that **EPD generalizes effectively beyond vision**, justifying the broader **“LMM”** terminology. We will make this clearer in the final version.
>
> ---
>
>
> ### **Q2: Perception as a Natural Extension of DistServe with added Encoding**
>
> Our system is inspired by DistServe’s disaggregation approach, but **disaggregating encoding from prefill** presents **unique challenges and opportunities** beyond prefill-decode separation.
>
> - First, we identify **Intra-Request Parallelism (IRP)** as crucial for reducing TTFT in encoding-heavy workloads. Otherwise, the benefits of disaggregation can be offset by communication overhead, since both E and P stages have similar characteristics (compute-bound).
> - Second, **optimized resource allocation** is essential, as different LMMs exhibit diverse compute profiles (e.g., LLaVA is prefill-heavy, while MiniCPM is encoding-heavy), requiring stage-specific tuning of resources.
> - Third, in dynamic workloads, growing **pipeline imbalance can nullify IRP gains**, making **dynamic role switching** vital to maintain high throughput and efficiency.
>
> These system-level insights are foundational to the design and performance gains demonstrated by EPD.
>
> ---
>
> Additionally, feel free to explore our released code at this anonymized [link](https://drive.google.com/drive/folders/1cEyGCPw54EkgBjZs73m-SZq51M2etD0X).
>
> ---
>
> ### **Q3: Formalism of Equation (2), concrete example, and Search Space**
>
> Here is an explanation and we will expand this in the appendix and include examples in the released code.
>
> Equation (2) is a **general formulation** for expressing optimization objectives in a disaggregated inference setting. It is intentionally designed to be flexible across deployment scenarios.
>
> It captures the trade-off between system performance (e.g., throughput or SLO attainment) and deployment cost, and is designed to flexibly adapt to different deployment scenarios. Below, we explain two practical real-world use cases:
>
> **1. Offline Optimization:**
> Consider a small business that wants to deploy with 8 GPUs, the user’s goal is to maximize **goodput** under a fixed hardware budget. In this case, the cost term in Equation (2) is constant (since all GPUs are active), and the objective simplifies to maximizing throughput. The user collects historical workload traces and feeds them to our resource allocation module. The optimizer searches over: (1) number of instances per stage (E/P/D), (2) degree of parallelism (IRP, TP, PP), (3) batch sizes, and (4) scheduling strategies. This yields the best static deployment setup for the given workload.
>
> **2. Online Serving with Autoscaling:**
> In dynamic cloud settings, Equation (2) incorporates both **performance (e.g., SLO attainment)** and **cost (e.g., number of active workers)**. The objective might be to balance **SLO attainment** (e.g., 95% of requests meeting latency targets) against **cost** (e.g., number of active GPU workers). The system collects request traces periodically and re-solves Equation (2) to adapt to changes in workload—for example, shifting from 3E2P3D to 2E1P5D when decoding becomes the bottleneck.
>
> ---

---

### Decision · Program_Chairs · 2025-05-01

**Decision:**

Accept (poster)

**Comment:**

The submission initially got three Weak Accepts. After the rebuttal discussion, one reviewer increased his score from Weak Accept to Accept, while the other two reviewers maintained their Weak Accepts but remain positive about the submission. After reading the reviews, I think the proposed approach EPD can bring interest in the community and can have practical benefit in industry. Because of these reasons, I recommend acceptance of the paper. I would just encourage the authors to polish the final paper with the feedback provided by the reviewers.